# Circulating miR-1246 Targeting UBE2C, TNNI3, TRAIP, UCHL1 Genes and Key Pathways as a Potential Biomarker for Lung Adenocarcinoma: Integrated Biological Network Analysis

**DOI:** 10.3390/jpm10040162

**Published:** 2020-10-11

**Authors:** Siyuan Huang, Yong-Kai Wei, Satyavani Kaliamurthi, Yanghui Cao, Asma Sindhoo Nangraj, Xin Sui, Dan Chu, Huan Wang, Dong-Qing Wei, Gilles H. Peslherbe, Gurudeeban Selvaraj, Jiang Shi

**Affiliations:** 1Department of Oncology, The First Affiliated Hospital of Zhengzhou University, No.1 Jianshe East Road, Zhengzhou 450052, China; huangsy1989822@163.com (S.H.); suixin0319@163.com (X.S.); 2College of Science, Henan University of Technology, Zhengzhou 450001, China; ykwei@haut.edu.cn; 3Centre for Research in Molecular Modeling and Department of Chemistry and Biochemistry, Concordia University, 7141 Sherbrooke Street West, Montréal, QC H4B 1R6, Canada; satyavani.mkk@haut.edu.cn (S.K.); dqwei@sjtu.edu.cn (D.-Q.W.); Gilles.Peslherbe@concordia.ca (G.H.P.); gurudeeb99@haut.edu.cn (G.S.); 4Center of Interdisciplinary Science-Computational Life Sciences, College of Biological Engineering, Henan University of Technology, No.100, Lianhua Street, Hi-Tech Development Zone, Zhengzhou 450001, China; 5Department of General Surgery, Henan Tumor Hospital, No.127 Dongming Road, Zhengzhou 450008, China; caoyanghui1010@163.com; 6The State Key Laboratory of Microbial Metabolism, College of Life Sciences and Biotechnology, Shanghai Jiao Tong University, Shanghai 200240, China; asma_sindhoo@sjtu.edu.cn; 7Department of Respiratory, The First Affiliated Hospital of Zhengzhou University, No.1 Jianshe East Road, Zhengzhou 450052, China; jiaoluodeguangmang@163.com (D.C.); wanghuanyisheng@163.com (H.W.)

**Keywords:** lung adenocarcinoma, circulating miR-1246, glycosaminoglycan binding, prognosis, PI3K–Akt signaling pathways, TargetScan, UBE2C

## Abstract

Analysis of circulating miRNAs (cmiRNAs) before surgical operation (BSO) and after the surgical operation (ASO) has been informative for lung adenocarcinoma (LUAD) diagnosis, progression, and outcomes of treatment. Thus, we performed a biological network analysis to identify the potential target genes (PTGs) of the overexpressed cmiRNA signatures from LUAD samples that had undergone surgical therapy. Differential expression (DE) analysis of microarray datasets, including cmiRNAs (GSE137140) and cmRNAs (GSE69732), was conducted using the Limma package. cmiR-1246 was predicted as a significantly upregulated cmiRNA of LUAD samples BSO and ASO. Then, 9802 miR-1246 target genes (TGs) were predicted using 12 TG prediction platforms (MiRWalk, miRDB, and TargetScan). Briefly, 425 highly expressed overlapping miRNA-1246 TGs were observed between the prediction platform and the cmiRNA dataset. ClueGO predicted cell projection morphogenesis, chemosensory behavior, and glycosaminoglycan binding, and the PI3K–Akt signaling pathways were enriched metabolic interactions regulating miRNA-1245 overlapping TGs in LUAD. Using 425 overlapping miR-1246 TGs, a protein–protein interaction network was constructed. Then, 12 PTGs of three different Walktrap modules were identified; among them, ubiquitin-conjugating enzyme E2C (UBE2C), troponin T1(TNNT1), T-cell receptor alpha locus interacting protein (TRAIP), and ubiquitin c-terminal hydrolase L1(UCHL1) were positively correlated with miR-1246, and the high expression of these genes was associated with better overall survival of LUAD. We conclude that PTGs of cmiRNA-1246 and key pathways, namely, ubiquitin-mediated proteolysis, glycosaminoglycan binding, the DNA metabolic process, and the PI3K–Akt–mTOR signaling pathway, the neurotrophin and cardiomyopathy signaling pathway, and the MAPK signaling pathway provide new insights on a noninvasive prognostic biomarker for LUAD.

## 1. Introduction

Resection-based therapy is a key player that increases the patient’s survival in nonsmall-cell lung cancer (NSCLC). However, long-term survival remains below 50% in NSCLC patients as there is a frequent recurrence of disease development following surgery and treatment [1]. It may increase many concerns associated with a favorable therapeutic strategy. Analysis of circulating microRNAs (cmiRNAs) before surgical operation (BSO) and after the surgical operation (ASO) has provided significant information for NSCLC diagnosis, progression, and outcomes of treatment [2]. One of the epigenetic biomarkers, known as cmiRNA, serves as a potential source for diagnosing NSCLC and its subtypes [3,4,5,6]. There are three important advantages in using cmiRNAs as a biomarker for NSCLC; they are as follows: (1) diagnostic feasibility from body fluids, (2) elevated stability and protection from endogenous enzymes (RNAase), and (3) accumulation of pathologic information from various tumorous sites, which overcomes the difficulty of tumor heterogenicity [7]. One of the common subtypes in NSCLC is lung adenocarcinoma (LUAD), which accounts for ~40% of all lung cancers. Stable and essential biomarkers for early diagnosis of LUAD are still insufficient [8]. Notably, the cmiRNAs are more stable in serum samples [9], it facilitates the augmentation of miRNA as a promising blood-based diagnostic biomarker. Aberrant changes in the level of miRNA, correlated with tumor growth, results in metastasis, invasion, drug resistance, and progression in LUAD patients. The expression level of miRNA-33a plays a vital role in the progression of LUAD; it could be an ideal biomarker for the diagnosis and prognosis of LUAD patients who have received adjuvant chemotherapy [10]. Upregulated oncogenic miRNAs (miR-130b, miR-182-5p, miRNA-17, and miRNA-222) were reported to cause the development and progression of LUAD [11,12,13]; moreover, downregulated miRNAs (miR-486-5p, miR-101, miR-133a), also called tumor-suppressive miRNAs, were reported to repress the development of NSCLC [14,15,16]. miR-21 and miR-24 were significantly lower in ASO serum samples of lung carcinoma patients when compared to the samples of BSO patients. The findings depicted that both miRNAs (21 and 24) could be employed as biomarkers for the prediction of cancerous growth reappearance ASO [2]. Besides that, Asakura et al. [17] compared the diagnostic indexes of miR (17-3p, -1268b, and -6075) BSO and ASO of serum samples of LC (*n* = 180) patients. After surgery, the level of miRNAs was significantly reduced; it enhances their potential in the screening of resectable lung cancer, including adenocarcinoma.

Gene signatures are an essential condition for potential clinical practice in cancer. It has many important inferences that are used to reclassify the disease [18]. The identification of gene signatures from patients who have undergone surgical therapy provides new insights on the diagnosis and therapeutic implications of LUAD. Thus, we performed integrated biological network analysis to identify the potential target genes (PTGs) of the shared cmiRNA signatures BSO and ASO of LUAD samples. Microarray analysis of gene expression profiles is a standard and well-known method to identify key hub genes and pathways [19,20]. Initially, we collected the cmiRNA and cmRNA datasets from the Gene Expression Omnibus (GEO) database. Then, we performed differential expression (DE) using the Limma algorithm and identified miR-1246 as a potential upregulated gene in BSO and ASO samples of LUAD. miR-1246 TGs were predicted from 12 different TG prediction platforms. Then, the overlapping genes of miR-1246 TGs and DE-cmRNAs were used to construct a protein–protein interaction (PPI) network. After that, different modules were extracted from the PPI network using the Walktrap algorithm. Moreover, 12 potential target genes (PTGs) were predicted from the modules based on degree centrality measures, and their functional and pathway enrichment terms were determined. Furthermore, the PTGs were validated using the expression of miR-1246 and the PTG correlation analysis survival curve and immune–histochemical analysis. This study will provide new insights into the underlying molecular mechanism in LUAD, which might contribute to the clinical therapy of LUAD patients.

## 2. Materials and Methods

### 2.1. Data Collection

Microarray datasets GSE137140 and GSE69732 of cmiRNAs and cmRNAs were extracted from the GEO database [17,21]. The workflow of the study is shown in Figure 1. The cmiRNA study was performed using a 3D-Gene Human miRNA V21_1.0.0 platform, which included cmiRNA profiles of 3924 samples consisting of 1566 BSO, 180 ASO of cancer, and 2178 noncancer controls. The histological types include adenocarcinoma (1217), squamous carcinoma (221), adenosquamous carcinoma (18), and small cell carcinoma (23) and other (87) subtype samples. However, in this study, we included 1217 BSO and 180 ASO LUAD samples and 1774 noncancer controls. The cmRNA study was performed using the Illumina HumanHT-12 WG-DASL V4.0 R2 expression bead chip platform, which included cmRNA profiles of six samples of lung cancer and noncancer controls.

### 2.2. Differential Expression of cmiRNAs and cmRNAs

The R package “limma” (linear models for microarray analysis) of the Bioconductor project was used to retrieve, preprocess datasets, and perform differential expression of cmiRNAs [22]. Log2 transformation, Benjamini and Hochberg and *t*-test were used to perform normalization and calculate the false discovery rate (FDR; *p* < 0.05) of samples [23]. The total number of samples were divided into two groups, control versus BSO samples and control versus ASO samples, for cmiRNAs and cmRNAs as LUAD versus control. The analysis demonstrated that miR-1246 was upregulated in both samples, which was used for further analysis.

### 2.3. miR-1246 Target Gene Prediction

The web interface miRWalk 2.0 (http://zmf.umm.uni-heidelberg.de/apps/zmf/mirwalk2/) was employed to predict the target genes of miR-1246. The database contains comprehensive and experimentally verified information about miRNA-target gene interaction [24]. We extracted all the possible miR-1246 target genes from the database. It includes 11 other different miRNA-target prediction resources, namely, TargetScan, miRanda, miRDB, mirbridge, miRmap, miRNAMap, PITA, Pictar2, RNA22, and RNAhybrid. The target genes of miR-1246 were predicted from more than five databases and were used for further analysis.

### 2.4. Screening of Overlapping Target Genes

The overlapping miR-1246 target genes between the list of genes predicted from 12 different databases and DE-cmRNAs in LUAD samples were selected based on the standard log2FC >1. A Venny (https://bioinfogp.cnb.csic.es/tools/venny/index.html) [25] intersection diagram was used to facilitate more distinctively identified miR-1246 TGs from more than five databases and DE-cmRNAs in LUAD samples. These overlapping genes were employed to perform sequential bioinformatics analysis to discover the molecular mechanism of miR-1246 in LUAD.

### 2.5. Construction of PPI Network

The search tool for retrieval of interacting genes/proteins (STRING; https://string-db.org/) [26] is a database that is used to construct the PPI network. Currently, the database consists of 18,838 human proteins with a core of 25,914,693 network interactions. In this study, we constructed the PPI network from identified TGs using the STRING interactome. The highest confidence interaction score was set to 0.9, which reduces false-positive interactions [27].

### 2.6. Identification of Modules and Hub Genes

The R package “*igraph*” was used to extract modules based on the Walktrap algorithm from the PPI network. It runs several short random walks within a group of nodes that are highly connected to detect small modules. From the modules, the PTGs (nodes) were identified with two different centrality measures, “degree” and “betweenness” [28]. The degree of the gene is the number of maximum connections it has with the other genes. Genes with a high degree act as hubs within the network. The betweenness of a gene is the number of paths that pass through it when considering the pair-wise shortest paths between all genes in the network. A node that occurs between two dense clusters will have a high betweenness.

### 2.7. Functional Enrichment Analysis

We have used ClueGO v2.5.3, which is a Cytoscape v3.8.0 plugin for function and pathway enrichment analysis of PTGs [29,30]. A list of overlapping miR-1246 TGs or PTGs was provided as input into ClueGO with selected specific parameters, such as species (*Homo sapiens*), ID type (Entrez gene ID), and different enrichment functions (biological process or cellular component or molecular function or KEGG pathways), for the analysis. Each enrichment was calculated based on the Bonferroni method (*p* < 0.005) and a kappa score of 0.96. In ClueGO, the kappa score is employed to identify term–term interactions revealed as edges on the network and correlate terms and pathways into functional groups, depending on shared genes. The high kappa score indicates stronger network connectivity of PTGs to the GO terms.
*k* = p_o_ − p_e_ / 1 − p_e_ = 1 − (1 − p_o_ /1 − p_e_)
where p_o_ represents a relative observed agreement among raters, and p_e_ represents the hypothetical probability of chance agreement [31]. Functional enrichment analysis results were visualized using ImageGP (http://www.ehbio.com/ImageGP/index.php/Home/Reg/reg.html).

### 2.8. Validation of Potential Target Genes (PTGs)

#### 2.8.1. Expression of PTGs in LUAD

The Gene Expression Profiling Interactive Analysis 2 (GEPIA2; http://gepia.cancer-pku.cn/) web interface was employed to validate the expression level of PTGs in LUAD [32]. GEPIA consists of comprehensive RNA sequencing information from TCGA and the Genotype–Tissue Expression (GTEx) project. The expression level of PTGs is illustrated in the box plot, and *p* < 0.05 was considered to be statistically important.

#### 2.8.2. Correlation Analysis of miR-1246 and PTGs

The expression data of miR-1246 and PTGs in LUAD were obtained from the TCGA database. The expression data were transformed with log2 for normalization. Then, the relationship with miR-1246 and PTGs was elucidated using Spearman’s correlation analysis. The linear regression plot was used as a visual representation of the trend of the relationships. It was performed with GraphPad Prism (USA).

#### 2.8.3. Survival Analysis

The PTGs were identified from the Walktrap modules. The R package “*survival*” was employed to calculate the Kaplan–Meier (KM) survival plot with a hazard ratio (HR) and log-rank test of the hub, which was implemented in the KM plotter web interface [33]. The database retrieved the gene expression profiles, and clinical data include TNM (Stage I, II, III, and IV), gender (male and female), smoking history (smoker and nonsmoker), histology (adenocarcinoma and squamous cell carcinoma), and grade (G1, G2, G3, and GX) of 1925 patients from The Cancer Genome Atlas (TCGA), Cancer Biomedical Informatics Grid (caBIG), and GEO. We analyzed the overall survival rate of the PTGs as input and obtained the plot from the tool.

#### 2.8.4. Protein Expression Analysis in LUAD

The Human Protein Atlas database (HPAD) was used to validate the immune-histochemistry of PTGs. The database facilitates system-level studies on the transcriptome of the coding genes and pathological expression of genes in different cancer types. The staining profiles for proteins of the PTGs in human LUAD tissue based on immunohistochemistry using tissue microarrays. Further, the name of the antibody, tissue type, staining levels (high, medium, low, and not detected), intensity, and quality of the IHC analysis data were retrieved from the database for interpreting results [34,35].

## 3. Results

### 3.1. Differentially Expressed cmiRNAs and cmRNAs

In total, 5132 DE-cmiRNAs, which included 2242 underexpressed and 324 overexpressed cmiRNAs, were obtained from BSO samples. Moreover, 1646 underexpressed and 920 overexpressed cmiRNAs were obtained from ASO samples. The top 10 overexpressed and underexpressed cmiRNAs are illustrated in Table 1. In both the samples, miR-1246 was highly upregulated, with log2FC as 7.09 in BSO samples and 6.28 in ASO samples. Therefore, further studies were carried out using miR-1246. There were 306 overexpressed cmRNAs, and 743 under-expressed cmRNAs identified from differential expression. The top 10 overexpressed and underexpressed cmRNAs are illustrated in Table 2.

### 3.2. Identification of Overlapping miR-1246 Target Genes

Briefly, 9802 miR-1246 TGs were predicted by 12 target gene prediction platforms. Additionally, the differential expression mRNA of LUAD predicted 1049 genes. The intersection of these gene sets using Venny demonstrated that 425 miR-1246 TGs were highly expressed in LUAD (Figure 2A).

### 3.3. Functional and Pathway Enrichment of Overlapping miR-1246 Target Genes

The functional and pathway enrichment terms have a great consequence in the regulatory mechanism of miR-1246 target genes. Plasma-membrane-bound cell projection morphogenesis, chemosensory behavior, and neuron development are important biological process terms (Figure 2B), and glycosaminoglycan binding (Figure 2C) is the major molecular functional term of miR-1245 overlapping target genes. KEGG pathways have demonstrated that nicotine addiction, neomycin, kanamycin and gentamicin biosynthesis, complement and coagulation cascades, ECM-receptor interaction, and PI3K–Akt signaling pathways are enriched pathway terms regulating miR-1245 overlapping target genes in LUAD (Figure 2D).

### 3.4. Modules and PTGs Identification

Using 425 overlapping miR-1246 TGs, the PPI network was constructed with 3133 nodes and 4228 interactions (Figure 3). Then, using the Walktrap algorithm, 21 modules with a minimum of three nodes were predicted from the PPI network. Among them, the top three highly interconnected modules, having more numbers of nodes, were selected for PTG analysis. Module 1 (33 nodes; *p* = 1.98 × 10^−74^); Module 2 (15 nodes; *p* =1.21 × 10^−08^), and Module 3 (10 nodes; *p* =1.62 × 10^−14^) were employed to identify the PTGs (Figure 4). Table 3 demonstrates the degree and betweenness centrality measures of 12 different PTGs, which include ubiquitin-conjugating enzyme E2C (UBE2C), tubulin folding cofactor E (TBCE), DnaJ heat shock protein family (Hsp40) member A3 (DNAJA3), paired like homeodomain 2 (PITX2), transforming growth factor-beta-induced factor 1 (TGIF1), T-cell receptor alpha locus interacting protein (TRAIP), ubiquitin c-terminal hydrolase L1 (UCHL1), troponin I3 (TNNI3), troponin T1 (TNNT1), neuroblastoma RAS (NRAS) viral oncogene, Rac family small GTPase 3 (RAC3), and the Ephrin-A4 (EFNA4) precursor.

### 3.5. Function and Pathway Enrichments of PTGs

The following were enriched molecular function terms: In Module 1, the TNF signaling pathway, ubiquitin-mediated proteolysis, NOD-like receptor signaling pathways; in Module 2, regulation of MAP kinase activity, negative regulation of cysteine-type endopeptidase activity involved in the apoptotic processes, and transmembrane receptor kinase activity; in Module 3, channel inhibitor activity, calcium-dependent ATPase activity, and calmodulin-binding. The top three biological processes of the three different modules are microtubule cytoskeleton organization, the DNA metabolic process, regulation of the intrinsic apoptotic signaling pathway, activation of MAPK activity, positive regulation of endothelial cell proliferation, Rac protein signal transduction, positive regulation of cell–matrix adhesion, regulation of the force of heart contraction, and regulation of skeletal muscle contraction, respectively. The figure indicates the significant BP terms in the three modules (Figure 5A). PTGs of modules were enriched in many signaling pathways, including necroptosis, protein processing in the endoplasmic reticulum, ubiquitin-mediated proteolysis, Parkinson’s disease, pathways in cancer, the MAPK signaling pathway, the mTOR signaling pathway, the Ras signaling pathway, the PI3K−Akt signaling pathway, Epstein−Barr virus infection, cardiac muscle contraction, adrenergic signaling in cardiomyocytes, hypertrophic cardiomyopathy, dilated cardiomyopathy, and the neurotrophin signaling pathway. Among the pathways, three pathways, namely, cancer, ubiquitin-mediated proteolysis, and Epstein−Barr virus infection, had a high number of gene counts (>75). Ten pathways had a moderate level of gene counts (>50), including the MAPK signaling pathway, the mTOR signaling pathway, the Ras signaling pathway, the PI3K−Akt signaling pathway, necroptosis, protein processing in the endoplasmic reticulum, and dilated cardiomyopathy; two pathways had fewer gene counts (>25) (Figure 5B).

### 3.6. Validation of PTGs

#### 3.6.1. Expression of PTGs

There were 12 different PTGs identified from the modules, namely, UBE2C, TBCE, DNAJA3, PITX2, TGIF1, TRAIP, UCHL1, TNNI3, TNNT1, NRAS, RAC3, and EFNA4, and they demonstrated a high level of expression in LUAD tissues (Figure 6). As miR-1246 was upregulated in LUAD, the differentially expressed genes in LUAD have essential importance to act as potential target genes of miR-1246.

#### 3.6.2. Spearman’s Correlation Analysis of PTGs

Spearman’s correlation analysis indicated that four of the 12 PTGs was significantly and positively correlated with miR-1246: UBE2C (*r* = 0.32, *p* = 2.2 × 10^−08^), TNNT1 (*r* = 0.023, *p* = 0.07), TRAIP (*r* = 0.58, *p* = 8.7 × 10^−28^), and UCHL1 (*r* = 0.44, *p* = 6.5 × 10^−15^) (Figure 7).

#### 3.6.3. Prognostic Impact of PTGs

KM plots demonstrated the prognostic impact of the PTGs, which was identified from three different modules of the PPI network. The results explained that the high expression of UBE2C, UCHL1, TRAIP, TNNT1, TNNI3, and RAC3 were associated with poor overall survival of lung adenocarcinoma patients (*p* < 0.05; Figure 8). Moreover, the high expressions of PITX2, NRAS, ENFA4, DNAJA3, TBCE, and TGIF1 were correlated with longer overall survival of LUAD patients (Figure 9).

#### 3.6.4. Protein Expression of PTGs

The immune–histochemistry of pathological slides of the human protein atlas database (HPAD) indicated that the protein expressions of PTGs were drastically higher in LUAD tissues compared with adjacent normal tissues (Figure 10). The IHC data for UBE2C (https://www.proteinatlas.org/ENSG00000175063-UBE2C/pathology/lung+cancer#img), UCHL1 (https://www.proteinatlas.org/ENSG00000154277-UCHL1/pathology/lung+cancer#img), TRAIP (https://www.proteinatlas.org/ENSG00000183763-TRAIP/pathology/lung+cancer#img), and RAC3 (https://www.proteinatlas.org/ENSG00000169750-RAC3/pathology/lung+cancer#img) have a strong intensity, and the intensity indicates that these PTGs played an initiative role and may be used as a biomarker. The other PTGs have weak or low intensity, which may be a reason for the availability of a limited number of samples in the database.

## 4. Discussion

Continuous intricacy in earlier diagnosis is the main reason for the increased rate of LUAD individuals. Discovering potential and novel biomarkers and its interactive gene-level mechanism may lead to higher chances in the diagnosis and prognosis of LUAD [36]. Prognostic markers that include growth factor and hormone receptors, proliferation and angiogenesis markers, and proteases provide molecular characteristics and assist the course of therapy [37,38,39]. On this concern, the result of the study highlighted miR-1246 gene targets and key signaling pathways in LUAD.

MiR-1246 plays an imperative role in different cancers through their targets. For example, the expression of miR-1246 was significantly correlated with chemoresistance and cancer stem-cell-like characteristics and could identify a worse prognosis in cancer patients (pancreatic) by targeting cyclin G2 protein-coding genes such as CCNG2 [40]. Moreover, Li et al. [41] reported that miR-1246 enhances the proliferation and invasion of breast cancer cells by repressing the level of its CCNG2 target gene. Du et al. [42] found that miR-1246 targeted thrombospondin-2 (THBS2) to inhibit cancerous growth and enhanced apoptosis in cervical cancer. Moreover, miR-1246 directly targeted death receptor 5 (DR5), which promotes proliferation and increases radioresistance in lung cancers [43]. However, to our knowledge, the specific role of miR-1246 in LUAD has been insufficiently investigated. Thus, in the present findings, we examined miR-1246 expression through miRNA sequencing data, which increased promisingly in LUAD patients. Hypothetically, target genes facilitate the functions of miR-1246. Therefore, we studied the probable target genes of miR-1246 and its enriched pathways through KEGG analysis.

For the biological process, we explored plasma-membrane-bound cell projection morphogenesis as a key function. The top five functional terms were chemosensory behavior, neuron development, negative regulation of neuron death, biomineral tissue development, and cellular component morphogenesis and axonogenesis. For molecular function, glycosaminoglycan binding is the major term. Glycosaminoglycan binds to different protein targets through electrostatic interactions between positively charged amino acids and negatively charged uronic acids. It is found to be concerned in multiple signaling cascades as it is mandatory for angiogenesis, cancer invasion, and metastasis. Similarly, it can also inhibit tumor progression and act as a drug target [44]. Salanti et al. [45] reported that parasite-derived protein could be exploited to target not only common but also complex malignancies like melanoma-associated glycosaminoglycan modification. Moreover, the targeting of glycosaminoglycans chains by tetrabranched peptide-like NT4 provides insights into the role of heparan sulfate proteoglycans in cancer cell adhesion and migration [46]. Based on those findings, we speculated that the regulation of miRNAs in glycosaminoglycan binding might improve the efficacy of LUAD therapeutics theoretically. Exclusively, we hypothesized that miR-1246 might be implicated in the regulation of glycosaminoglycans, which may persuade the treatment of LUAD patients.

Accordingly, we constructed a PPI interaction network of the TGs and selected the most densely connected modules based on the degree and between centrality measures. Moreover, we studied the gene expressions of UBE2C, TBCE, DNAJA3, PITX2, TGIF1, TRAIP, UCHL1, TNNI3, TNNT1, NRAS, RAC3, and EFNA4, which were high in LUAD tissue samples. Among these, UBE2C, TNNT1, TRAIP, and UCHL1 were positively correlated with miR-1246. The protein expression of ENFA4, NRAS, RAC3, TRAIP, UBE2C, and UCHL1 were upregulated in LUAD compared with that of control.

On survival analysis, we determined that UBE2C, UCHL1, TRAIP, TNNT1, TNNI3, and RAC3 were associated with poor overall survival of LUAD patients. Moreover, the high expression of PITX2, NRAS, ENFA4, DNAJA3, TBCE, and TGIF1 was correlated with longer overall survival of LUAD patients. The high expression of UBE2C is found in the advanced stage of cancer, which might point out its involvement in cancer progression and invasion. Additionally, patients with higher UBE2C levels showed a shorter overall survival (OS) time and worst OS prognosis. It indicated that UBE2C overexpression positively correlated in several cancers [47]. Accordingly, we hypothesized that post-translational modification of protein like UBE2C plays key roles in protein degradation, and protein interactions and their dysregulations in the earlier stage may lead to LUAD. Based on that, miR-1246 can be implied to target UBE2C, which may contribute to improving LUAD patients’ prognosis and the survival of LUAD patients.

The ubiquitin-conjugating enzyme-2C (UBE-2C) is majorly responsible for the destructive cleavage of mitotic cyclin proteins for spindle assembly, which leads to the progression of the cell cycle. Moreover, the expressions of UBE-2C protein or mRNA are aberrantly expressed in various cancer types that lead to poor clinical results. Therefore, UBE-2C acts as a potential biomarker in cancer [48]. In the case of gastric cancer, Zhang et al. [49] reported that upregulated miR-17/20a significantly enhances the growth of gastric cancer cells by directly targeted UBE-2C. Jin et al. [50] reported that the miR-548e-5p, together with UBE-2C and zinc finger E-box binding homeobox (ZEB1/2), acts as a potential diagnostic biomarker and target for NSCLC. Moreover, another ubiquitin-protein UCHL1 was reported to promote uterine serous cancer cell proliferation, cell cycle progression [51], and TGFβ-induced breast cancer metastasis [52]. So far, no study has mentioned the relationship between miR-1246 and UBE-2C and UCHL1 in LUAD. We are the first to report the targeting relationship between miR-1246 and UBE-2C. Based on the key enriched GO items, we supposed that miR-1246 targets UBE-2C and UCHL1 for the regulation of ubiquitin-mediated proteolysis and, therefore, persuades the prognosis of LUAD patients. However, more evidence is needed to validate this hypothesis.

TNNT1 is one of the isoforms of the troponin protein and is highly expressed in skeletal muscle. It plays key roles in muscle contraction and relaxation. In addition, reports have suggested that TNNT1 could contribute to cell proliferation in breast cancer [53]. Moreover, Hao et al. [54] reported that TNNT1 might promote the progression of colon adenocarcinoma by mediating the epithelial–mesenchymal transition process. In the present study, we found that TNNT1 was involved in cardiac muscle contraction, regulation of skeletal muscle contraction, troponin 1, and C binding molecular functions and associated with poor prognosis of LUAD. As a result, we assumed that miR-1246 might target TNNT1 and, thus, be associated with skeletal muscle regulation and poor prognosis of LUAD patients. However, future studies are required to supply more evidence.

TRAIP is a ring-type E3 ubiquitin ligase involved in many cellular functions, namely, NF-κB activation, DNA damage response, mitosis, and carcinogenesis [55]. Initially, it was considered a tumor suppressor in basal cell carcinomas and breast cancer [56]. Moreover, Guo et al. [57] reported that TRAIP exhibited as an oncogene in liver cancer. In the present study, we found that TRAIP was involved in pathways in cancer, regulation of autophagy, and DNA metabolic process enrichment terms and associated with poor prognosis of LUAD. As a result, we assumed that miR-1246 might target TRAIP as the metabolic function plays a key role in the prognosis of LUAD patients. However, future studies are required to supply more evidence.

Moreover, we also expected to determine the pathways that are associated with miR-1246 and its potential target genes in LUAD. We identified 12 important pathways that might have an essential role in the incidence and development of LUAD: pathways in cancer, ubiquitin-mediated proteolysis, protein processing in the endoplasmic reticulum, necroptosis, hypertrophic cardiomyopathy, cardiac muscle contraction, focal adhesion, the PI3K–Akt and Ras signaling pathways, the neurotrophin and mTOR signaling pathways, and the MAPK signaling pathway. Many studies have reported that the PI3K–Akt signaling pathway, the Ras signaling pathway, the neurotrophic-signaling pathway, the mTOR-signaling pathway, and the MAPK signaling pathway are regulated in the development of lung adenocarcinoma patients [58,59]. The PI3K/AKT/mTOR signal pathway is a key intracellular signal transduction pathway, with an essential function in cell proliferation, growth, survival, vesicle trafficking, glucose transport, and cytoskeletal organization. As we mentioned above, downregulation of PITX2, TGIF1, and TRAIP, and upregulation of TBCE may be involved in the different cellular processes (transcription, RNA splicing, cell cycle, and apoptosis) through the PI3K–Akt–mTOR signaling pathway and the MAPK signaling pathway, which might improve the survival of LUAD patients. Additionally, underexpression of TNNI3, TNNT1, and PITX2 might involve four independent pathways of cardiac muscle contraction, adrenergic signaling in cardiomyocytes, hypertrophic cardiomyopathy, and dilated cardiomyopathy, which are also incidentally associated through some somatic mutation and proto-oncogenic activities in LUAD. Besides, targeting Epstein−Barr virus infection and necroptosis pathways regulates the development of LUAD and the nonapoptotic form of regulated cell death, which may advance the prognosis of LUAD patients. Moreover, ubiquitin-mediated proteolysis and Parkinson’s disease pathways are also indirectly associated through proteolysis and cell proliferation in LUAD. UBE-2C and UCHL1 are enriched in these pathways; we wondered if miR-1246 might target UBE-2C and UCHL1 and participate in the regulation of the cellular process, which may improve the prognosis of LUAD patients.

## 5. Conclusions

We have done a detailed and complete study about LUAD and miR-1246 using public datasets, with comprehensive biological network analysis useful for cancer research. Further experimental studies are still necessary to validate the results, which is a tough but promising task. We assumed that miR-1246 might target UBE2C, TNNT1, TRAIP, and UCHL1 during the regulation of ubiquitin-mediated proteolysis, glycosaminoglycan binding, DNA metabolism, the PI3K–Akt–mTOR signaling pathway, the neurotrophin and cardiomyopathy signaling pathway, and the MAPK signaling pathway. Upregulated UBE-2C, TNNT1, TRAIP, and UCHL1 may point out better survival of LUAD patients through the ubiquitin-mediated proteolysis, protein processing in the endoplasmic reticulum, and skeletal muscle contraction pathways. Moreover, similarly, this study had the limitations of other data-mining methods; the results of Limma and miRNA target prediction databases can be biased due to insufficient resources. To enhance the reliability of the results, immunohistochemical data from HPAD were employed for confirmation. Due to the constraint of HPAD, we could not get all the related IHCs of the tumor and adjacent normal samples of each potential target gene.

## Figures and Tables

**Figure 1 jpm-10-00162-f001:**
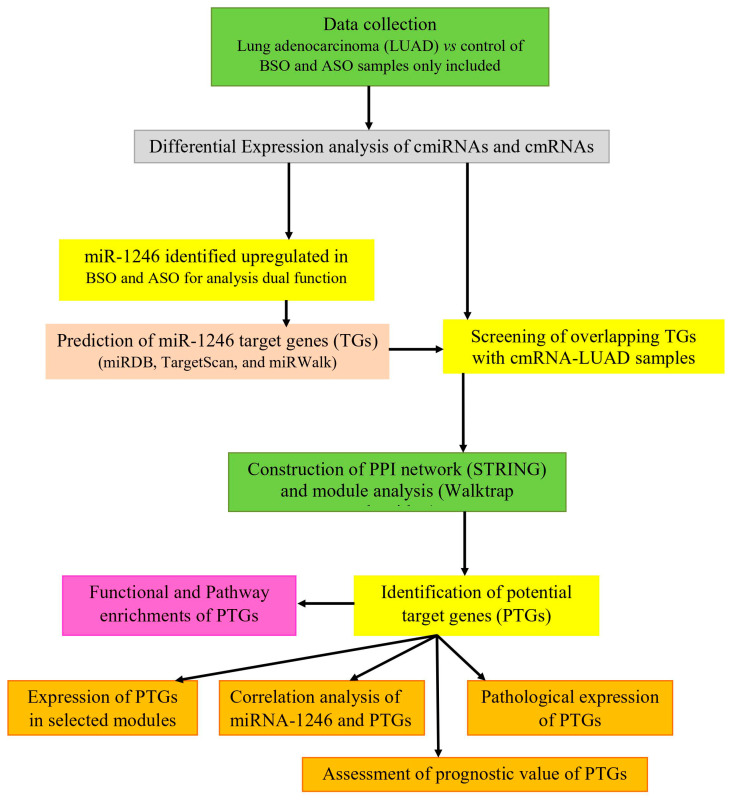
Schematic illustration of the study.

**Figure 2 jpm-10-00162-f002:**
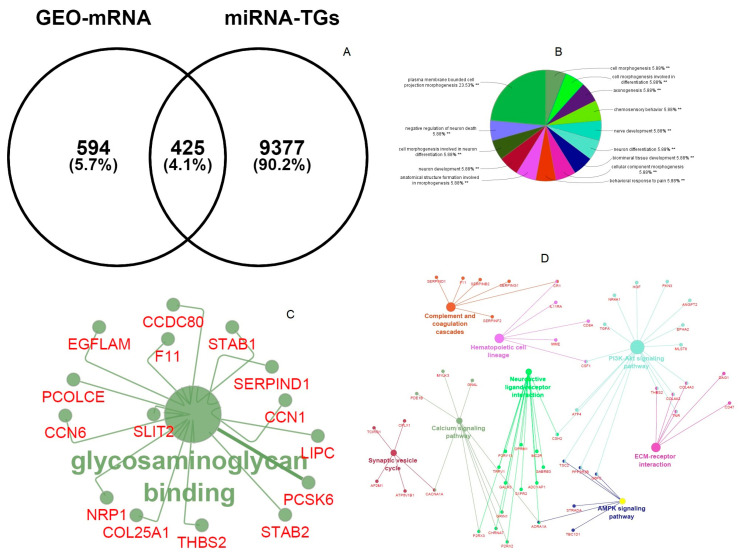
(**A**) Venny interactive diagram of overlapping miR-1246 genes and differential expression (DE)-cmRNAs. (**B**) Biological process of overlapping genes. (**C**) Molecular functions of overlapping genes. (**D**) KEGG pathway enrichment of overlapping genes.

**Figure 3 jpm-10-00162-f003:**
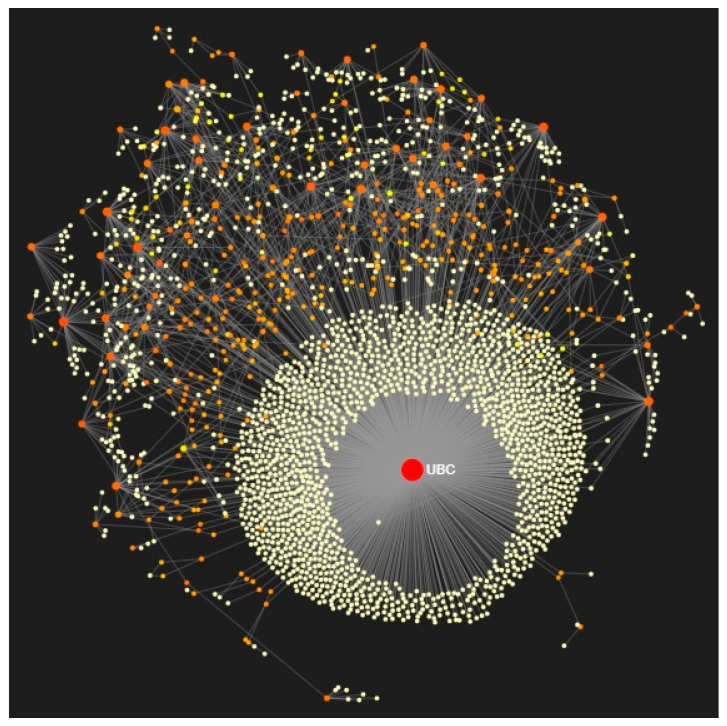
Protein–protein interaction (PPI) network of overlapping genes of miR-1246 targets and differentially expressed circulating mRNAs in force atlas layout (red color indicates downregulated genes, orange color indicates upregulated genes, and yellow color indicates interconnected genes). Change in the size of the nodes depends on degree centrality measures. UBC is the major node of the subnetwork that is enriched in ubiquitin-mediated proteolysis.

**Figure 4 jpm-10-00162-f004:**
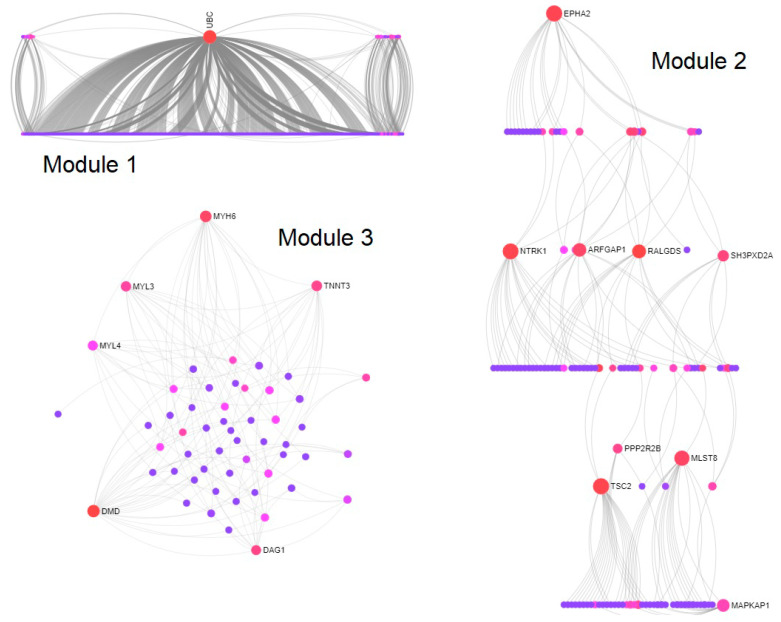
Walktrap modules of potential target genes extracted from the PPI network. Sugiyama layout of Module 1 (33 nodes; *p* = 1.98 × 10^−74^); linear bipartite/tripartite layout of Module 2 (15 nodes; *p* = 1.21 × 10^−08^); circular bipartite/tripartite layout of Module 3 (10 nodes; *p* = 1.62 × 10^−14^). Red color indicates downregulated, pink color indicates upregulated genes, and blue color indicates interconnected genes. Change in the size of the nodes depends on degree centrality measures.

**Figure 5 jpm-10-00162-f005:**
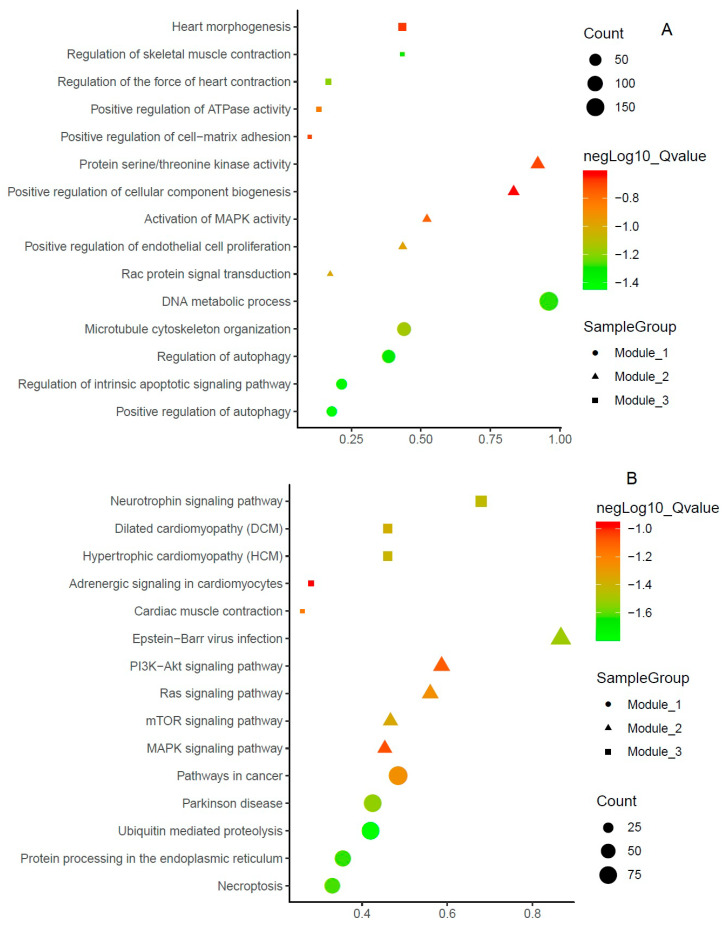
Functional enrichment terms of potential target genes. (**A**) Biological process; (**B**) KEGG pathways.

**Figure 6 jpm-10-00162-f006:**
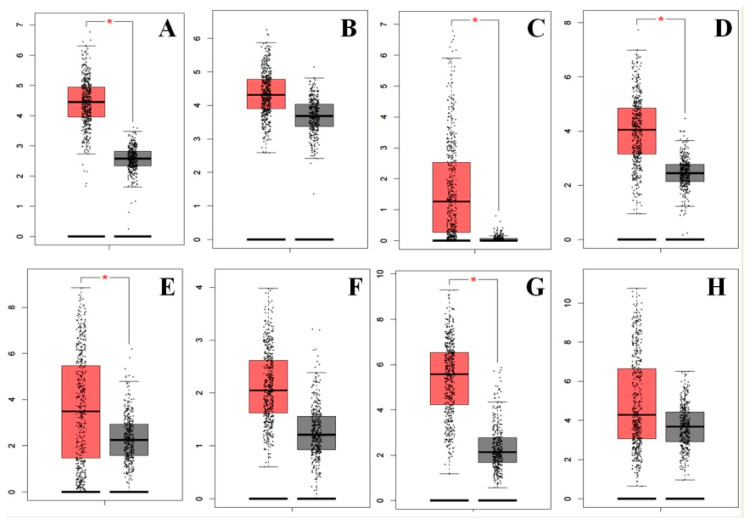
Expression of potential target genes in lung adenocarcinoma (LUAD) and control samples from the Gene Expression Profiling Interactive Analysis (GEPIA) database. Expression of the potential target genes was detected in 483 LUAD tissues (red in color) and 347 normal tissues (black in color). Eight potential target genes (**A**) EFNA4, (**B**) NRAS, (**C**) PITX2, (**D**) RAC3, (**E**) TNNT1, (**F**) TRAIP, (**G**) UBE2C, and (**H**) UCHL1 were upregulated in LUAD tissues compared to control.

**Figure 7 jpm-10-00162-f007:**
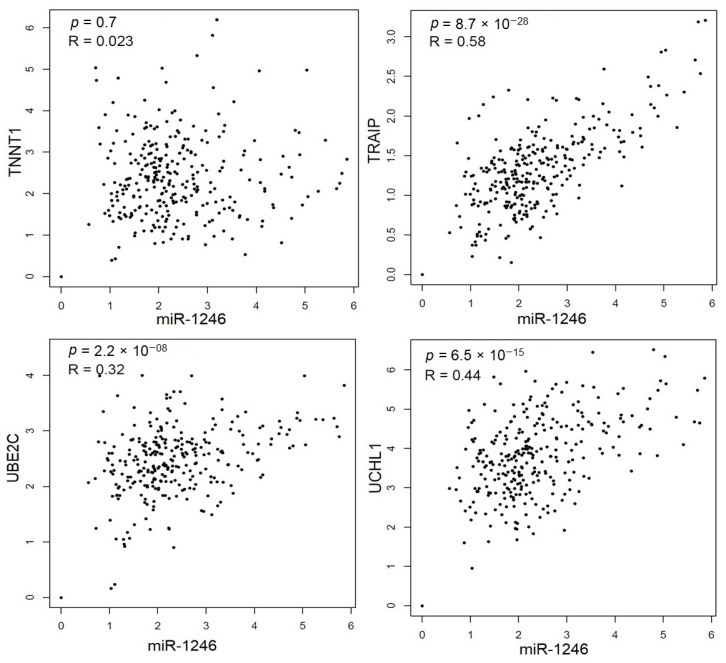
Correlation analysis between miR-1246 and 4 PTGs: UBE2C (*r* = 0.32, *p* = 2.2 × 10^−08^), TNNT1 (*r* = 0.023, *p* = 0.07), TRAIP (*r* = 0.58, *p* = 8.7 × 10^−28^), and UCHL1 (*r* = 0.44, *p* = 6.5 × 10^−15^).

**Figure 8 jpm-10-00162-f008:**
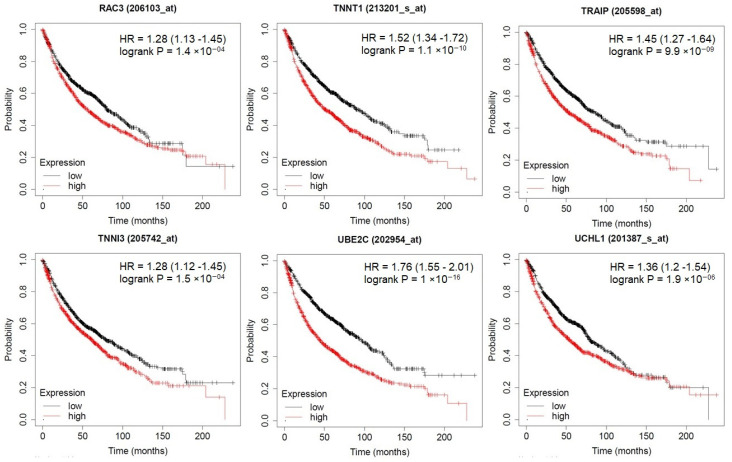
The prognostic value of PTGs in LUAD patients. High expression of UBE2C, UCHL1, TRAIP, TNNT1, TNNI3, and RAC3 was associated with poor overall survival of LUAD patients (*p* < 0.05).

**Figure 9 jpm-10-00162-f009:**
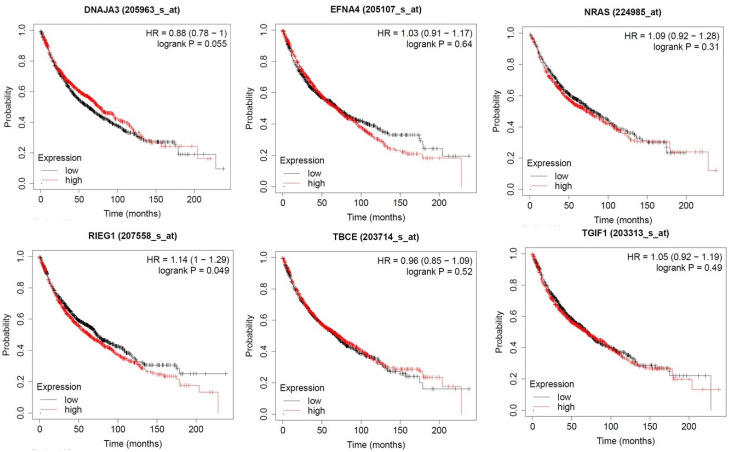
The prognostic value of PTGs in LUAD patients. High expression of PITX2, NRAS, ENFA4, DNAJA3, TBCE, and TGIF1 was correlated with longer overall survival of LUAD patients (*p* < 0.05).

**Figure 10 jpm-10-00162-f010:**
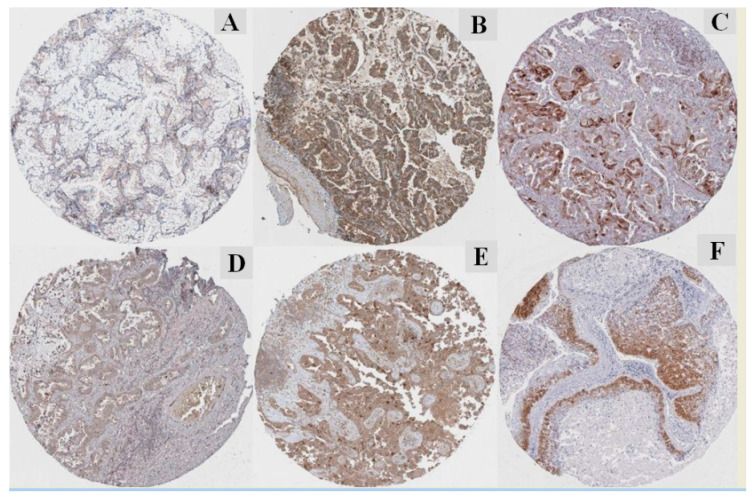
Immunohistochemistry of the PTGs based on the Human Protein Atlas database (HPAD). (**A**) Median staining of EFNA4 in LUAD (antibody: CAB021350; magnification of 4 × 10; substructures: cytoplasmic/membranous). (**B**) High staining of NRAS in LUAD (antibody: CAB010157; magnification of 4 × 10; substructures: cytoplasmic/membranous). (**C**) Medium staining of RAC3 in LUAD (antibody: HPA047820; magnification of 4 × 10; substructures: cytoplasmic/membranous). (**D**) Medium staining of TRAIP in LUAD (antibody: HPA036262; magnification of 4 × 10; substructures: cytoplasmic/membranous). (**E**) Medium staining of UBE2C in LUAD (antibody: CAB011464; magnification of 4 × 10; substructures: cytoplasmic/membranous. (**F**) Medium staining of UCHL1 in LUAD (antibody: CAB002580; magnification of 4 × 10; substructures: cytoplasmic/membranous). The staining intensity is strong and quantity: >75% to all the selected protein morphology.

**Table 1 jpm-10-00162-t001:** Top 10 overexpressed and underexpressed circulating miRNAs (cmiRNAs).

miRNA_ID	Log2FC	*p*-Value	miRNA_ID	Log2FC	*p*-Value
**BSO overexpressed**			**BSO underexpressed**		
hsa-miR-1246	6.28	2.79 × 10^−110^	hsa-miR-373-5p	−5.92	0
hsa-miR-8060	5.69	6.62 × 10^−189^	hsa-miR-1199-5p	−6.05	0
hsa-miR-920	5.46	0	hsa-miR-208b-5p	−6.07	0
hsa-miR-6131	5.31	9.32 × 10^−187^	hsa-miR-6777-5p	−6.07	0
hsa-miR-4259	5.08	9.10 × 10^−249^	hsa-miR-4648	−6.32	0
hsa-miR-6849-5p	4.61	2.22 × 10^−172^	hsa-miR-4435	−6.38	0
hsa-miR-193a-5p	4.39	4.87 × 10^−182^	hsa-miR-4276	−6.46	0
hsa-miR-6717-5p	4.24	2.02 × 10^−226^	hsa-miR-6857-5p	−6.49	0
hsa-miR-3934-5p	4.11	2.63 × 10^−128^	hsa-miR-92a-2-5p	−7.19	0
hsa-miR-1343-3p	3.96	0	hsa-miR-1203	−7.37	0
**ASO overexpressed**			**ASO underexpressed**		
hsa-miR-1246	7.09	0	hsa-miR-3184-5p	−8.41	0
hsa-miR-1290	6.17	0	hsa-miR-1203	−1.54	2.73 × 10^−214^
hsa-miR-29b-1-5p	6.03	0	hsa-miR-4730	−1.60	0
hsa-miR-191-5p	5.75	0	hsa-miR-873-3p	−1.64	1.79 × 10^−173^
hsa-miR-451a	5.64	0	hsa-miR-92a-2-5p	−1.74	0
hsa-miR-103a-3p	5.17	0	hsa-miR-4276	−1.89	2.65 × 10^−242^
hsa-miR-4755-3p	5.09	0	hsa-miR-3184-5p	−2.01	0
hsa-miR-6131	4.99	0	hsa-miR-4648	−2.05	3.64 × 10^−225^
hsa-miR-4771	4.96	0	hsa-miR-6857-5p	−2.36	4.82 × 10^−302^
hsa-miR-4480	4.89	0	hsa-miR-4481	−2.55	1.76 × 10^−312^

**Table 2 jpm-10-00162-t002:** Top 10 overexpressed and underexpressed cmRNAs.

Gene Symbol	Description	Log2FC	*p*-Value
**Overexpressed genes**			
BTBD11	BTB domain containing 11	3.108	4.69 × 10^−4^
ZNF683	Zinc finger protein 683	1.991	6.82 × 10^−3^
GPATCH4	G-patch domain containing 4	1.754	8.86 × 10^−4^
EHMT1	Euchromatic histone lysine methyltransferase 1	1.652	3.61 × 10^−3^
RAB6B	Ras-related protein Rab-6B	1.576	9.06 × 10^−3^
C12orf5	TP53 induced glycolysis regulatory phosphatase	1.569	1.44 × 10^−3^
GNLY	Granulysin	1.569	9.71 × 10^−3^
RPGRIP1	X-linked retinitis pigmentosa GTPase regulator-interacting protein 1	1.542	3.44 × 10^−4^
CPT1B	Carnitine palmitoyltransferase I	1.527	4.17 × 10^−3^
SRI	Sorcin	1.525	1.38 × 10^−3^
**Underexpressed genes**			
WISP3	WNT1-inducible-signaling pathway protein 3	−1.855	5.39 × 10^−3^
HFE2	Hemojuvelin	−1.858	3.08 × 10^−3^
LOR	Loricrin	−1.861	4.96 × 10^−3^
SLC26A11	Sodium-independent sulfate anion transporter	−1.875	3.97 × 10^−3^
DCAF12L2	DDB1- and CUL4-associated factor 12-like protein 2	−1.885	3.31 × 10^−4^
DKFZp564N2472	POM121 transmembrane nucleoporin-like 12	−1.885	4.22 × 10^−3^
FRG2C	FSHD region gene 2 family member C	−1.921	4.13 × 10^−4^
PRM2	Protamine 2	−1.95	8.97 × 10^−3^
PTCH2	Patched 2	−2.022	4.04 × 10^−3^
NNAT	Neuronatin	−2.298	9.95 × 10^−3^

**Table 3 jpm-10-00162-t003:** Degree and betweenness centrality measures of 12 different potential target genes (PTGs).

Official Symbol	Gene ID	Official Full Name	Chromosome Location	Exon Count	Degree	Betweenness
UBE2C	11,065	Ubiquitin conjugating enzyme E2 C	20q13.12	8	34	7811.25
TBCE	6905	Tubulin folding cofactor E	1q42.3	18	13	39.11
DNAJA3	9093	DNAJ heat shock protein family (Hsp40) member 3	16p13.3	12	12	6127.74
PITX2	5308	Paired-like homeodomain transcription factor 2	4q25	9	07	4584.14
TGIF1	7050	TGFB induced factor homeobox 1	18p11.31	12	07	22.32
TRAIP	10,293	TRAF interacting protein	3p21.31	16	06	1533.11
UCHL1	7345	Ubiquitin C-terminal hydrolase L1	4p13	9	06	1537.48
TNNI3	7137	Troponin I3	19q13.42	8	04	0.23
TNNT1	7138	Troponin T1	19q13.42	15	04	10.91
NRAS	4893	Neuroblastoma RAS viral oncogene homolog	1p13.2	7	03	247.07
RAC3	5881	Rac family small GTPase 3	17q25.3	6	03	630.68
EFNA4	1945	Ephrin A4	1q21.3	4	03	0

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
