# Peer review of "Circulating miR-1246 Targeting UBE2C, TNNI3, TRAIP, UCHL1 Genes and Key Pathways as a Potential Biomarker for Lung Adenocarcinoma: Integrated Biological Network Analysis"

_jpm, 2020, doi:10.3390/jpm10040162_

Round 1
Reviewer 1 Report
This manuscript is written for analysis of circulating miRNAs (cmiRNA) for lung adenocarcinoma (LUAD). Overall, it is a very well organized and very interesting paper. But "Introduction" is too long and Table 1-3 and Figure 3 (A or B) are not provided. The authors should exchange for clearer images (for example, Figure 2 and 4). In line 239, what dose "Figure" indicate? In page 13, the authors showed immunohistochemistry of the human protein atlas database (HPAD). This is a good understanding of protein expression of target genes. Was there a morphological difference between gene expression and adenocarcinoma?
Author Response
Reviewer 1
- This manuscript is written for analysis of circulating miRNAs (cmiRNA) for lung adenocarcinoma (LUAD). Overall, it is a very well organized and very interesting paper. However, "Introduction" is too long and Table 1-3 and Figure 3 (A or B) are not provided.
Thanks for your appreciation. As per reviewer suggestion, authors reduced the content of the introduction section. Besides, Table 1-3 are provided and typo on Figure 3 (A) and (B) corrected as figure 3 and figure 4. Revision are highlighted in yellow in color.
- The authors should exchange for clearer images (for example, Figure 2 and 4).
Thanks for your comment. The resolution of the Figure 2 and 4 is increased as per journal standard (300dpi)
- In line 239, what dose "Figure" indicate?
Thanks for your comment. The typo is corrected and the line removed “The complete function was illustrated in the Figure”.
- In page 13, the authors showed immunohistochemistry of the human protein atlas database (HPAD). This is a good understanding of protein expression of target genes. Was there a morphological difference between gene expression and adenocarcinoma?
Thanks for your valid question. The gene expression signature differentiates histology and further morphological observation by IHC confirmed the potential use (diagnosis/therapeutic) of target genes.
Reviewer 2 Report
The authors present interesting and useful data about LUAD and miR-1246using the public datasets with comprehensive biological network analysis. They conclude that PTGs of 40 cmiRNA-1246 and key pathways namely ubiquitin-mediated proteolysis, glycosaminoglycan 41 binding, DNA metabolic process, and PI3K-Akt signaling pathways provide new insights on the 42 non-invasive prognostic bio-marker for LUAD.
The manuscript is generally well written and results are clearly presented; however, a few issues need to be clarified:
Line 58: One of the subtypes in NSCLC is lung adenocarcinoma (LUAD) which accounts for ~40% of lung 59 cancers.
Lines 100-101 (data collection): The histological types include adenocarcinoma (1217), squamous carcinoma (221), adenosquamous carcinoma (18), and small cell carcinoma (23) and others (87) subtypes samples
Lines 303-304: LUAD is the chief and widespread type of NSCLC, which records for ~40% of lung malignancies.
It is not clear throughout the manuscript which exact subtypes of lung carcinoma were analysed in this study. According to lines 101-101 all subtypes were included. If this is the case, please correct the terminology (LUAD and NSCLC) since small cell carcinoma should be designed as SCLC. Moreover, if SCLC is included in the study, authors should differentiate their results and findings according to the two categories NSCLC vs SCLC since their prognosis and biological behaviour are extremely different and cnnot be accomunated.
If only LUAD were effectively included in the study please correct the terminology throughout the manuscript (Methods and discussion). Authors should also motivate their choice of including only adenocarcinomas in their analysis.
Author Response
Reviewer 2
The authors present interesting and useful data about LUAD and miR-1246using the public datasets with comprehensive biological network analysis. They conclude that PTGs of cmiRNA-1246 and key pathways namely ubiquitin-mediated proteolysis, glycosaminoglycan binding, DNA metabolic process, and PI3K-Akt signaling pathways provide new insights on the non-invasive prognostic bio-marker for LUAD. The manuscript is generally well written and results are clearly presented; however, a few issues need to be clarified:
Thanks for your appreciation.
- Line 58: One of the subtypes in NSCLC is lung adenocarcinoma (LUAD) which accounts for ~40% of lung 59 cancers.
Thanks for your comment. The line revised, as “One of the common subtypes in NSCLC is lung adenocarcinoma (LUAD) which accounts for ~ 40% of all lung cancers”.
- Lines 100-101 (data collection): The histological types include adenocarcinoma (1217), squamous carcinoma (221), adenosquamous carcinoma (18), and small cell carcinoma (23) and others (87) subtypes samples
Thanks for your comment. The details of dataset defined clearly as follows
The dataset contains totally 3924 samples, which consists of 1566 BSO, 180 ASO of cancer, and 2178 non-cancer controls. The histological types include adenocarcinoma (1217), squamous carcinoma (221), adenosquamous carcinoma (18), and small cell carcinoma (23) and others (87) subtypes samples. However, in this study, we have included 1217 (BSO), 180 (ASO) of LUAD samples, and 1774 non-cancer controls.
- Lines 303-304: LUAD is the chief and widespread type of NSCLC, which records for ~40% of lung malignancies.
Thanks for your comment. The repeated line “LUAD is the chief and widespread type of NSCLC, which records for ~40% of lung malignancies” is removed.
- It is not clear throughout the manuscript which exact subtypes of lung carcinoma were analysed in this study. According to lines 101-101 all subtypes were included. If this is the case, please correct the terminology (LUAD and NSCLC) since small cell carcinoma should be designed as SCLC. Moreover, if SCLC is included in the study, authors should differentiate their results, findings according to the two categories NSCLC vs SCLC since their prognosis and biological behaviour are extremely different, and cnnot be accomunated. If only LUAD were effectively included in the study please correct the terminology throughout the manuscript (Methods and discussion). Authors should also motivate their choice of including only adenocarcinomas in their analysis.
Thanks for your comment. Authors used LUAD and their adjacent control samples throughout the study. The content revised in methods as follows
The dataset contains totally 3924 samples, which consists of 1566 BSO, 180 ASO of cancer, and 2178 non-cancer controls. The histological types include adenocarcinoma (1217), squamous carcinoma (221), adenosquamous carcinoma (18), and small cell carcinoma (23) and others (87) subtypes samples. However, in this study, we have included 1217 (BSO), 180 (ASO) of LUAD samples, and 1774 non-cancer controls.
The terminology is corrected as LUAD throughout the manuscript in the appropriate places.
Reviewer 3 Report
The authors Huang present the manuscript "Circulating miR-1246 targeting UBE2C, TNNI3, TRAIP, UCHL1 genes, and key pathways as a potential biomarker for lung adenocarcinoma: Integrated biological network analysis"
This is an interesting paper with potential for further research. The authors are delved into a complex analysis carried out with the datasets available. However, as with many manuscripts, there are usually items that require attention to bring the paper to a quality that is required for peer review by the journal for publication.
There are several items listed below in range from minor and major points for the author's consideration and corrections.
- Figure 1: Resolution of this image should be improved
- There are major and numerous typographical and formatting errors throughout the whole paper, with double spaces, combined words with symbols i.e. line 157 “wherespo” error type occurs frequently. As well as falsely combined words which should have spaces between them. There are too numerous to list all here. Please review the whole document and correct formatting.
- MDPI provides the use of English editing services here at https://www.mdpi.com/authors/english
- The version of Cytoscape is not provided nor is a reference to the platform, particularly since used in conjunction with ClueGO
- Line 203-204 does not scan as a complete sentence. Please revise.
- Figure 3 caption requires further explanation, as it stands this image means very little. What to do the various coloured nodes mean? We see Red hub node UBC in the centre, it is not defined or described in the caption. The primary interacting partners subsidiary to it are not annotated. The subsequent yellow, orange and dark orange nodes are not described. No significance is denoted to the change in the size of the nodes, or if this has any relevance?
- Figure 3B does not exist in the document it is mentioned in line 221
- Table 3 does not exist in the document and is mentioned in line 221
- The descriptors of module 1 and 2 are not adequately described and have defined relationship to any of the figures within this manuscript until the final line in section 3.5. This needs to be clarified and defined earlier in the paragraph
- Figure 4 A and B requires resolution correction, it has been distorted by increasing the size of a small image
- There are no methodologies included for the use of immunohistochemistry slides used for figure 9. What does the colour of the stains mean? Which substructures are stained with different dyes? What are the different dyes?
- Figure 9 caption needs to relay the above points information as well
- The results and discussion sections read as though they were compiled and written by different people, there are a disjointedness and no flow of ideas between the sections. Some work needs to be done here to make this manuscript more cohesive in its overall presentation.
- The content used in the discussion is appropriate with sufficient references it, however, reads like a list of compiled dot points
- The use of personal pronouns should be avoided wherever possible in scientific writing to remain objective.
- Lines 339-342 requires rewording; it scans as an incomplete sentence.
- Lines 390-400 report PI3K-Akt, mTOR and MAPK signalling as regulated in cancer, however, this is not described how it is changed. These signalling pathways are intricately interwoven in many cellular cycles and functionality processes in normal, developing or disease type cell/s. What is the impact to your study and analysis in finding the links between these and the other 9 listed pathways?
- The abstract and keywords tell the readers that there is in-depth analysis and discussion about PI3K-Akt as it is mentioned twice in the abstract and keyword. However, there is only a single mention in the results and only briefly mentioned in the closing paragraph of the discussion.
- Ambiguity in the conclusion section needs to be amended
- Line 405-406 requires rewording, misleading adjectives used.
- Line 413-416 needs to remove the hyperbole in the final statement.
Author Response
Reviewer 3
The authors Huang present the manuscript "Circulating miR-1246 targeting UBE2C, TNNI3, TRAIP, UCHL1 genes, and key pathways as a potential biomarker for lung adenocarcinoma: Integrated biological network analysis". This is an interesting paper with potential for further research. The authors are delved into a complex analysis carried out with the datasets available. However, as with many manuscripts, there are usually items that require attention to bring the paper to a quality that is required for peer review by the journal for publication. There are several items listed below in range from minor and major points for the author's consideration and corrections.
Thanks for your appreciation.
- Figure 1: Resolution of this image should be improved
Thanks for your comment. The resolution of Figure 1 is increased as per journal standard (300dpi)
- There are major and numerous typographical and formatting errors throughout the whole paper, with double spaces, combined words with symbols i.e. line 157 “wherespo” error type occurs frequently. As well as falsely combined words which should have spaces between them. There are too numerous to list all here. Please review the whole document and correct formatting.
Thanks for your comment. The typos, falsely combined words, and spaces are corrected in the revised version as per suggestion.
- MDPI provides the use of English editing services here at https://www.mdpi.com/authors/english
Thanks for your comment. Yes, the authors used MDPI service for English editing to the revised version.
- The version of Cytoscape is not provided nor is a reference to the platform, particularly since used in conjunction with ClueGO
Thanks for your comment. The content revised as follows “We have used ClueGO v2.5.3, which is a Cytoscape v3.8.0 plugin for function and pathway enrichment analysis of PTGs”
- Line 203-204 does not scan as a complete sentence. Please revise.
Thanks for your comment. The content revised as follows “The functional and pathway enrichment terms were having a great consequence in the regulatory mechanism of miR-1246 target genes”.
- Figure 3 caption requires further explanation, as it stands this image means very little. What to do the various coloured nodes mean? We see Red hub node UBC in the centre, it is not defined or described in the caption. The primary interacting partners subsidiary to it are not annotated. The subsequent yellow, orange and dark orange nodes are not described. No significance is denoted to the change in the size of the nodes, or if this has any relevance?
Figure 3. PPI network of overlapping genes of miR-1246 targets and differentially expressed circulating mRNAs in force atlas layout (red color indicates downregulated and orange color indicates for upregulated genes and yellow color indicates interconnected genes). Change in the size of the nodes depends on the degree centrality measures. UBC is the major node of the subnetwork that is enriched in ubiquitin-mediated proteolysis.
- Figure 3B does not exist in the document it is mentioned in line 221
Thanks for your comment. Figure 3B included as figure 4 in the revised form.
Figure 4. Walktrap modules of potential target genes extracted from PPI network. Sugiyama layout of module 1 (33 nodes; p =1.98×10-74); linear bipartite/tripartite layout of module 2 (15 nodes; p =1.21×10-08) and circular bipartite/tripartite layout module 3 (10 nodes; p =1.62×10-14). Red color indicates downregulated, pink color indicates for upregulated genes, and blue color indicates interconnected genes). Change in the size of the nodes depends on the degree centrality measures.
- Table 3 does not exist in the document and is mentioned in line 221
Thanks for your comment. Table 3 included in the revision.
- The descriptors of module 1 and 2 are not adequately described and have defined relationship to any of the figures within this manuscript until the final line in section 3.5. This needs to be clarified and defined earlier in the paragraph
Thanks for your comment. The details of the modules highlighted in Figure 4. Revisions are highlighted in yellow in color. The potential target genes from the respective modules described in Table 3. The functional enrichment terms of each module explained in section 3.5. as follow
In module 1, TNF signaling pathway, ubiquitin-mediated proteolysis, NOD-like receptor signaling pathways; in module 2, regulation of MAP kinase activity, negative regulation of cysteine-type endopeptidase activity involved in the apoptotic processes, and transmembrane receptor kinase activity and in module 3, channel inhibitor activity, calcium-dependent ATPase activity, and calmodulin-binding were enriched molecular function terms. The top three biological process of three different modules are microtubule cytoskeleton organization, DNA metabolic process, regulation of intrinsic apoptotic signaling pathway, activation of MAPK activity, positive regulation of endothelial cell proliferation, Rac protein signal transduction, positive regulation of cell-matrix adhesion, regulation of the force of heart contraction and regulation of skeletal muscle contraction respectively. The figure indicates the significant BP terms in three modules (Figure 5(A)). PTGs of modules were enriched in many signaling pathways includes Necroptosis, protein processing in the endoplasmic reticulum, Ubiquitin mediated proteolysis, Parkinson disease, Pathways in cancer, MAPK signaling pathway, mTOR signaling pathway, Ras signaling pathway, PI3K−Akt signaling pathway, Epstein−Barr virus infection, cardiac muscle contraction, adrenergic signaling in cardiomyocytes, hypertrophic cardiomyopathy, dilated cardiomyopathy and neurotrophin signaling pathway. Among the pathways, three pathways namely cancer, Ubiquitin mediated proteolysis and Epstein−Barr virus infection had a high number of gene counts (>75). Ten pathways had a moderate level of gene counts (>50) includes MAPK signaling pathway, mTOR signaling pathway, Ras signaling pathway, PI3K−Akt signaling pathway, necroptosis, protein processing in the endoplasmic reticulum, and dilated cardiomyopathy (DCM), and two pathways having fewer gene counts (>25) (Figure 5(B)).
- Figure 4 A and B requires resolution correction, it has been distorted by increasing the size of a small image
Thanks for your comment. The resolution of Figure 4A and 4B is increased as per journal standard (300dpi)
- There are no methodologies included for the use of immunohistochemistry slides used for figure 9. What does the colour of the stains mean? Which substructures are stained with different dyes? What are the different dyes?
Thanks for your comment. The details are included in the methods and figure legends as follows
2.8.4. Protein expression analysis in LUAD
The Human Protein Atlas database (HPAD) was used to validate the immune-histochemistry of PTGs. The database facilitates system-level studies on the transcriptome of the coding genes and pathological expression of genes in different cancer types. The staining profiles for proteins of the PTGs in human LUAD tissue based on immunohistochemistry using tissue microarrays. Further, the name of the antibody, tissue type, staining levels (high, medium, low, and not detected), intensity, and quality of IHC analysis data were retrieved from the database [34-35].
Figure 9. Immuno-histochemistry of the PTGs based on the HPAD. Median staining of EFNA4 in LUAD (antibody: CAB021350; magnification of 4×10; substructures: Cytoplasmic/membranous); (B)High staining of NRAS in LUAD (antibody: CAB010157; magnification of 4×10; substructures: Cytoplasmic/membranous); (C) Medium staining of RAC3 in LUAD (antibody: HPA047820; magnification of 4×10; substructures: Cytoplasmic/membranous); (D) Medium staining of TRAIP in LUAD (antibody: HPA036262; magnification of 4×10; substructures: Cytoplasmic/membranous); (E) Medium staining of UBE2C in LUAD (antibody: CAB011464; magnification of 4×10; substructures: Cytoplasmic/membranous); (F) Medium staining of UCHL1 in LUAD (antibody:CAB002580; magnification of 4×10; substructures: Cytoplasmic/membranous). The staining intensity is strong and quantity: >75% to all the selected protein morphology.
- Figure 9 caption needs to relay the above points information as well
Thanks for your comment. The details are included in the figure legends as follows
Figure 9. Immuno-histochemistry of the PTGs based on the HPAD. Median staining of EFNA4 in LUAD (antibody: CAB021350; magnification of 4×10; substructures: Cytoplasmic/membranous); (B)High staining of NRAS in LUAD (antibody: CAB010157; magnification of 4×10; substructures: Cytoplasmic/membranous); (C) Medium staining of RAC3 in LUAD (antibody: HPA047820; magnification of 4×10; substructures: Cytoplasmic/membranous); (D) Medium staining of TRAIP in LUAD (antibody: HPA036262; magnification of 4×10; substructures: Cytoplasmic/membranous); (E) Medium staining of UBE2C in LUAD (antibody: CAB011464; magnification of 4×10; substructures: Cytoplasmic/membranous); (F) Medium staining of UCHL1 in LUAD (antibody:CAB002580; magnification of 4×10; substructures: Cytoplasmic/membranous). The staining intensity is strong and quantity: >75% to all the selected protein morphology.
- The results and discussion sections read as though they were compiled and written by different people, there are a disjointedness and no flow of ideas between the sections. Some work needs to be done here to make this manuscript more cohesive in its overall presentation.
Thanks for your comment. The results and discussion are revised as per suggestion.
- The content used in the discussion is appropriate with sufficient references it, however, reads like a list of compiled dot points
Thanks for your comment. The discussion is improved the way of scientific style through editing in the revised form.
- The use of personal pronouns should be avoided wherever possible in scientific writing to remain objective.
Thanks for your comment. As per suggestion, the usage of personal pronouns is reduced in the revised form.
- Lines 339-342 requires rewording; it scans as an incomplete sentence.
Thanks for your comment. The sentence is corrected in the revised form.
Besides, we studied that the gene expression of UBE2C, TBCE, DNAJA3, PITX2, TGIF1, TRAIP, UCHL1, TNNI3, TNNT1, NRAS, RAC3, and EFNA4 were high in LUAD tissue samples.
- Lines 390-400 report PI3K-Akt, mTOR and MAPK signalling as regulated in cancer, however, this is not described how it is changed. These signalling pathways are intricately interwoven in many cellular cycles and functionality processes in normal, developing or disease type cell/s. What is the impact to your study and analysis in finding the links between these and the other 9 listed pathways?
Thanks for your valid comment. As per the reviewer's suggestion, the roles of pathways in LUAD described briefly in the discussion.
Many studies reported that the PI3K-Akt signaling pathway, Ras signaling pathway, neurotrophin signaling pathway, mTOR signaling pathway, and MAPK signaling pathway regulated in the development of lung adenocarcinoma patients [58-59]. PI3K/AKT/mTOR signal pathway is a key intracellular signal transduction pathway with an essential function in cell proliferation, growth, survival, vesicle trafficking, glucose transport, and cytoskeletal organization. As we mentioned above, down-regulation of PITX2, TGIF1, TGIF1, TRAIP, and upregulation of TBCE may be involved in the different cellular process (transcription, RNA splicing, cell cycle, and apoptosis) through PI3K-Akt -mTOR signaling pathways and MAPK signaling pathway which might improve the survival of LUAD patients. Also, under-expression of TNNI3, TNNT1, PITX2 might involve four independent pathways cardiac muscle contraction, adrenergic signaling in cardiomyocytes, hypertrophic cardiomyopathy, and dilated cardiomyopathy are also incidentally associated through some somatic mutation and proto-oncogenic activities in LUAD. Besides, targeting Epstein−Barr virus infection and necroptosis pathways regulates the development of LUAD and non-apoptotic form of regulated cell death, which may advance the prognosis of LUAD patients. Moreover, ubiquitin-mediated proteolysis; Parkinson's disease pathways are also indirectly associated through proteolysis and cell proliferation in LUAD. UBE2C and UCHL1 are enriched in these pathways; we wondered that miR-1246 might target UBE2C, UCHL1, and participate in the regulation of the cellular process, which may improve the prognosis of LUAD patients.
- The abstract and keywords tell the readers that there is in-depth analysis and discussion about PI3K-Akt as it is mentioned twice in the abstract and keyword. However, there is only a single mention in the results and only briefly mentioned in the closing paragraph of the discussion.
Thanks for your valid comment. As per reviewer's suggestion, the pathway information in the abstract and discussion is revised.
We conclude that PTGs of cmiRNA-1246 and key pathways namely ubiquitin-mediated proteolysis, glycosaminoglycan binding, DNA metabolic process, and PI3K-Akt-mTOR signaling pathway, neurotrophin, and cardiomyopathy signaling pathway, and MAPK signaling pathway provide new insights on the non-invasive prognostic biomarker for LUAD.
- Ambiguity in the conclusion section needs to be amended
Thanks for your valid comment. As per reviewer's suggestion, the conclusion amended.
We assumed that miR-1246 might target UBE2C, TNNT1, TRAIP, and UCHL1 during the regulation of ubiquitin-mediated proteolysis, glycosaminoglycan binding, DNA metabolism, PI3K-Akt-mTOR signaling pathway, neurotrophin and cardiomyopathy signaling pathway, and MAPK signaling pathway.
- Line 405-406 requires rewording, misleading adjectives used.
Thanks for your valid comment. As per reviewer's suggestion, the statement revised.
We assumed that miR-1246 might target UBE2C, TNNT1, TRAIP, and UCHL1 during the regulation of ubiquitin-mediated proteolysis, glycosaminoglycan binding, DNA metabolism, PI3K-Akt-mTOR signaling pathway, neurotrophin and cardiomyopathy signaling pathway, and MAPK signaling pathway.
- Line 413-416 needs to remove the hyperbole in the final statement.
Thanks for your valid comment. As per reviewer's suggestion, Line 413-416 is removed.